# The Workplace and Psychosocial Experiences of Australian Senior Doctors during the COVID-19 Pandemic: A Qualitative Study

**DOI:** 10.3390/ijerph19053079

**Published:** 2022-03-05

**Authors:** Jonathan Tran, Karen Willis, Margaret Kay, Kathryn Hutt, Natasha Smallwood

**Affiliations:** 1The Melbourne Medical School, The University of Melbourne, Parkville, VIC 3010, Australia; jbtran@student.unimelb.edu.au; 2Public Health, College of Health and Biomedicine, Victoria University, Footscray Park, Melbourne, VIC 3011, Australia; karen.willis@vu.edu.au; 3Division of Critical Care and Investigative Services, Royal Melbourne Hospital, Grattan Street, Parkville, VIC 3050, Australia; 4General Practice Clinical Unit, Level 8 Health Sciences Building, Royal Brisbane and Women’s Hospital, Herston, QLD 4029, Australia; m.kay1@uq.edu.au; 5Doctors’ Health Advisory Service, Suite 207, 69 Christie Street, St Leonards, NSW 2065, Australia; kathryn.hutt@dhas.org.au; 6Department of Respiratory Medicine, The Alfred Hospital, Prahan, VIC 3004, Australia; 7Department of Allergy, Immunology and Respiratory Medicine, Central Clinical School, The Alfred Hospital, Monash University, Melbourne, VIC 3004, Australia

**Keywords:** coronavirus, COVID-19, healthcare workers, frontline, senior doctors, psychosocial, mental health

## Abstract

The coronavirus disease 2019 (COVID-19) pandemic has had significant mental health impacts among healthcare workers (HCWs), related to resource scarcity, risky work environments, and poor supports. Understanding the unique challenges experienced by senior doctors and identifying strategies for support will assist doctors facing such crises into the future. A cross-sectional, national, online survey was conducted during the second wave of the Australian COVID-19 pandemic. Inductive content analysis was used to examine data reporting workplace and psychosocial impacts of the pandemic. Of 9518 responses, 1083 senior doctors responded to one or more free-text questions. Of the senior doctors, 752 were women and 973 resided in Victoria. Four themes were identified: (1) work-life challenges; (2) poor workplace safety, support, and culture; (3) poor political leadership, planning and support; and (4) media and community responses. Key issues impacting mental health included supporting staff wellbeing, moral injury related to poorer quality patient care, feeling unheard and undervalued within the workplace, and pandemic ill-preparedness. Senior doctors desired better crisis preparedness, HCW representation, greater leadership, and accessible, authentic psychological wellbeing support services from workplace organisations and government. The pandemic has had significant impacts on senior doctors. The sustainability of the healthcare system requires interventions designed to protect workforce wellbeing.

## 1. Introduction

It is well recognised that healthcare workers (HCWs) globally experience higher levels of work-related stress and mental illness compared to the general population [1,2,3,4]. Poor mental health outcomes in HCWs are related to heavy workloads, long working hours, high expectations and demands, poor work–life balance, and the emotional challenges of caring for sick and dying patients [5,6]. Doctors in particular are at increased risk of burnout and stress when compared to other occupational groups [7,8,9], with 10% of doctors in Australia reporting previous thoughts of suicide [10].

During crises such as the current coronavirus disease 2019 (COVID-19) pandemic, healthcare services and the health workforce are placed under enormous stress [11,12] with increased workload, altered work practices (including but not limited to adoption of telehealth and use of personal protective equipment (PPE), role changes (such as redeployment to another medical team), inadequate healthcare resources, large volumes of new information, changing health policies and guidelines, and greater risk of infection with COVID-19 [13,14,15,16,17]. Additionally, population-wide challenges have included lockdowns, social restrictions, loss of non-essential services, and reduced income [16,18,19].

Australian and international studies examining the impacts of the COVID-19 pandemic have reported increased fatigue, distress, anxiety, depression, and burnout amongst frontline HCWs [20,21,22]. While quantitative psychosocial effects of the pandemic on HCWs have been described [20,23,24], there has been very little qualitative research examining HCWs’ lived experiences and expectations related to crisis management [14,16]. The Australian COVID-19 Frontline Healthcare Workers Study examined the severity, prevalence, and predictors of mental health issues in Australian HCWs during the pandemic [20]. This article focuses on a sub-set of the data that reports the experiences of senior doctors to provide an understanding of their perspectives and lived experiences of the workplace and psychosocial impacts of the COVID-19 pandemic. Senior doctors have many years of clinical experience, undertake various workplace responsibilities (as clinicians, educators, and leaders), and have additional personal and family commitments. Given their experience and position in the health hierarchy, their experiences as both leaders and clinicians mean they are uniquely placed to deepen our understanding regarding the impacts of the pandemic. In this sub-study we aimed to identify the experiences and challenges faced by senior doctors during the COVID-19 pandemic and draw on their recommendations regarding workplace wellbeing and future crisis preparedness.

## 2. Materials and Methods

### 2.1. Australian COVID-19 Frontline Healthcare Workers Study

The full study methodology has been previously published [20]. To summarise, after ethics approval, a cross-sectional, anonymous, national, online survey was conducted during the second wave of the Australian COVID-19 pandemic between the 27th of August and 23rd of October 2020. Survey information was emailed to hospital leaders throughout Victoria and around Australia for dissemination to colleagues. The survey was also shared by 36 professional colleges, societies and associartions, university medical schools, and government health departments across Australia. Participants were those who self-identified as ‘frontline’ HCWs. They were not required to have cared for patients with COVID-19 to participate in the survey. There were no incentives to participate.

All participants provided online consent to participate in this study. Participants completed the online survey via a direct link or via a purpose-built website, and data were collected using REDcap electronic data capture tools [25]. Data collected in the survey questionnaire included: demographics, home life, professional background, work arrangements and changes, workplace leadership, and symptoms of mental illness (both subjectively determined and objectively measured using five validated psychological scales). In addition, four optional free-text questions were included regarding the challenges of the pandemic on mental health and more broadly (Table 1).

### 2.2. Senior Doctors Sub-Study

Only responses to the free text questions from participants who identified as senior medical staff working in secondary and tertiary care were included in this sub-study. Senior medical staff refers to senior doctors who have completed postgraduate specialist training and examinations, and who are overall in charge of a patient’s care. Senior doctors are often referred to as consultants or specialists in many countries. Responses from general practitioners are not included in this sub-study. These have been analysed separately due to their different experiences within the primary care sector. Senior doctors who did not provide any responses to the free-text questions were excluded from this qualitative analysis. An inductive qualitative content approach was used to analyse the free-text responses. This approach aims to describe patterns in the data and interpret meaning from the content [26,27]. The responses to each question were first thoroughly read by JT for data immersion. Based on initial reading, a codebook was developed categorising data into three types of responses: personal issues, workplace issues, or broader societal issues. Each response was coded by JT, and an iterative process was undertaken with additions or amendments to the codebook to ensure emerging ideas were coded.

Data were coded until saturation was reached, meaning that no new ideas were emerging from the data [28]. Rigour in ensuring saturation was achieved by coding all responses by participants (*n* = 716) who had answered three or four of the free-text questions. Subsequently, every 20th response from those who had only answered one or two questions was coded to verify saturation. The remaining responses (*n* = 351) were not coded as saturation was obtained. Data codes, ideas, and themes were discussed by three researchers (JT, KW and NS) at weekly meetings, with the final themes reached through consensus discussion.

### 2.3. Ethics

Approval for this study was obtained through the Royal Melbourne Hospital Human Research Ethics Committee (HREC/67074/MH-2020).

## 3. Findings

### 3.1. Study Participants

Of 9518 survey participants, 7846 complete responses were received, including 1278 from senior doctors who formed the sample for this sub-study. Free-text responses to at least one of four questions were received from 1083 senior doctors (84.7%) (Table 1). One hundred and ninety-five senior doctors did not provide any responses to the free-text questions. Of the senior doctors, 96.2% were between the ages of 31–64 years, 58.8% were female, 76.1% resided in the state of Victoria, and 86% had more than 10 years of post-graduate experience (Table 2).

### 3.2. Overview of Findings

Senior doctors described stress, frustration, and uncertainty during the pandemic, which impacted their mental health. Many senior doctors reported disruptions to their roles and responsibilities at home and in the workplace, leading to substantial strain on their work–life balance and to burnout. Participants also reported feeling their safety was neglected at the workplace, and many felt undervalued by their organisation and the government, with poor leadership at all levels. Community factors, including media misinformation and the views and behaviour of the general public, also contributed to senior doctors feeling frustrated and unappreciated. Four major themes emerged from the data: (1) work-life challenges; (2) poor workplace safety, support, and culture; (3) poor political leadership, planning, and support; and (4) media and community responses (Figure 1).

### 3.3. Theme 1: Work-Life Challenges

Senior medical staff described distress, exhaustion, and burnout associated with work-life challenges and ongoing pandemic uncertainty (Table 3). Disruptions to personal life during the pandemic included a lack of self-care opportunities, inadequate social supports, and financial loss. Many senior doctors also identified the need for more time off to improve physical and mental wellbeing. Female senior doctors also raised issues of gender inequality and described distress related to increased family caring responsibilities, including new homeschooling roles. Senior doctors also felt overwhelmed with increased occupational responsibilities, including additional clinical duties and academic roles. They reported experiencing increased concern and distress associated with supporting the wellbeing of other healthcare workers, such as junior medical staff (JMS), with concerns about the progression of JMS through training during the pandemic. They also wrote about issues relating to quality of patient care associated with the need to rationalise and justify medical interventions, the reduction in access to usual health services, and the adoption of telehealth consultations.

### 3.4. Theme 2: Poor Workplace Safety, Support, and Culture

Senior doctors reflected on the ongoing issues in the healthcare system including inadequate resources, insufficient workplace support, and poor organisational leadership and communication, which were exacerbated by the COVID-19 pandemic (Table 4). Many participants reported workplace safety as a major challenge during the pandemic due to limited access to PPE, poor quality PPE, inadequate PPE training and guidelines, and unsafe working environments with inadequate space for social distancing and poor ventilation. Senior doctors felt at risk of becoming infected with COVID-19, with significant concerns about transmission to patients, staff, and family members. Senior doctors also believed that workplace leadership and communication during the pandemic were poor, and wrote about feeling unsupported, unheard, and undervalued. Many participants described ongoing mental health stigma in the workplace. Stigma was associated with concerns about consequences to employment and limited access to mental health support services. Worsening mental health coupled with a lack of psychological support raised concerns regarding future workforce retention among senior doctors.

### 3.5. Theme 3: Poor Political Leadership, Planning, and Support

Senior doctors described how government departments responded poorly to the increasing COVID-19 case numbers, which included victim blaming of infected HCWs and the slow development and implementation of policies and guidelines (Table 5). They argued that the health system was ill-prepared for this crisis and stressed the importance of developing disaster response plans. To improve future responses, senior doctors described the need for better government planning, a centralised department of health, and better representation of HCWs in organisations and government to facilitate medical policy making.

### 3.6. Theme 4: Media and Community Responses

Senior doctors were frustrated by the large amount of media misinformation, non-adherence to public health directives by the public, and disruptions within the community (Table 6). Poor efforts from the community to maintain safety left many senior doctors feeling frustrated, concerned for their own health, and underappreciated for the risks they took as HCWs during the pandemic.

## 4. Discussion

To our knowledge, this is the first article describing the in-depth psychological and workplace experiences of Australian senior doctors during the COVID-19 pandemic. Senior doctors represented a unique population with expertise in areas involving workflow and policies in the healthcare system. They provided important perspectives on psychosocial stressors related to workplace issues as hospital leaders and clinicians, as well as on broader societal issues as community members. Key factors that impacted psychological wellbeing in senior doctors included: work–life imbalance related to increased personal and occupational responsibilities, an unsafe and unsupportive workplace, poor political leadership and planning, and poor responses from the community and media. The findings highlight the need to actively support the wellbeing of senior doctors during crises, as a crucial aspect of preparedness planning, enabling workforce retention and the preservation of high-quality patient care. This study underscores the need to destigmatise mental illness and consider innovative approaches to psychological support, as well as strategies to improve access and uptake of these measures long term.

The experiences reported by senior doctors demonstrate broad-ranging personal and workplace impacts of the pandemic, contributing to worsening mental health. Similarly, a recent study of senior specialist doctors in Ireland reported burnout in 77% of doctors, with 64% experiencing adverse effects on their mental health from the COVID-19 pandemic [29]. Another study of mental health amongst obstetricians and gynaecologists in the United Kingdom (UK) during the COVID-19 pandemic showed high rates of major depressive disorder and generalised anxiety disorder in both junior and senior doctors [30]. These findings highlight the importance of identifying strategies that proactively address these concerns.

In addition to increased workplace challenges, senior doctors described many broad stressors during the pandemic. These were often associated with pandemic uncertainty, lockdown restrictions, social isolation, loss of supports, and financial disruption. Other qualitative studies examining psychosocial impacts of the COVID-19 pandemic in frontline healthcare workers in Australia [16], Canada [31], and the UK [32] identified similar personal, professional, and social stressors. These factors have also been recognised in the general adult population, with a recent study demonstrating an association between negative adverse effects on daily life (e.g., loss of work and unemployment) from COVID-19 lockdown restrictions and symptoms of depression or anxiety [33]. Therefore, some of the broader challenges reported by senior doctors in this study, were reflective of population-wide experiences.

Unique to senior doctors is their vital role as leaders in the healthcare workforce. This leadership role involves managing and caring for others—both staff and patients. Senior doctors expressed particular concern for junior doctors, including the impacts on career progression and disruption to training and education. Concerns about maintaining wellbeing and morale of more junior doctors during the COVID-19 pandemic has been reported previously in a qualitative study with doctors in Pakistan, the UK, and the United States of America [34]. These findings showcase the care senior doctors express toward their junior colleagues, which can further add to their emotional load in the workplace.

Concerns regarding the impact of the COVID-19 pandemic on the quality of patient care were highlighted by senior doctors. Having concerns related to the inadequate delivery of patient care while feeling powerless to change the system can be associated with ‘moral injury’, which is an emotional response that occurs following events that contravene a person’s moral values [35]. Moral injury related to patient care and resource scarcity have been identified in HCWs in Australia [14,23] and internationally [36,37,38] during the pandemic, and can contribute to the development of post-traumatic stress, depression and suicide [39]. Conversely, feeling appreciated by the community can be protective against moral injury and adverse mental health outcomes [23]. Importantly, some senior doctors in the current study reported feeling unappreciated by workplace leaders and government. The potential for this to exacerbate the experiences of moral injury and worsen doctors’ mental health is concerning.

Workplace safety, resource scarcity, and poor organisational support were key issues experienced by HCWs [40]. International studies on doctors, nurses, paramedics, allied health professionals, and management staff during the pandemic reported common concerns, which have contributed to mental health symptoms including stress, hypervigilance, fatigue, sleeping difficulties, and difficulties concentrating [41,42]. In addition, reports on HCWs in the National Health Service in the UK have linked feeling undermined and undervalued by their organisation and government with increased burnout, and reduced wellbeing and resilience [43]. As highlighted in our study, this is particularly important for senior doctors who represent leaders in the workplace. Despite their seniority in the medical team hierarchy, many participants still felt that their concerns—namely those regarding safety and provision of PPE—went unheard, leading to feeling undervalued by their organisation. This suggests that even in positions of relative “power”, senior doctors can struggle to implement change in the workplace. Therefore, in support of doctors’ health and wellbeing, the Royal Australian and New Zealand College of Psychiatrists have called for increased access to appropriate PPE for doctors as it represents both a public health and a mental health issue [44]. Similarly, the World Health Professions Alliance called upon governments to prioritise support for frontline HCWs during the COVID-19 crisis [45]. These findings suggest widespread issues regarding organisational and political leadership may have contributed to worsening mental health in senior doctors during the pandemic.

Barriers to accessing care for mental illness in doctors can have long-term mental health and wellbeing sequelae. There have been a range of barriers described including confidentiality concerns, embarrassment, and insufficient time [46]. Nevertheless, the stigma of mental illness, both broadly and in healthcare, remains a significant barrier to accessing mental health support for many doctors [46,47,48]. This stigma is often related to concerns regarding detrimental impacts on career progression and future employment [49], as well as concerns that disclosing mental illness could lead to mandatory reporting, and restrictions on their ability to practise medicine by the Australian Health Practitioner Regulation Agency [50]. To combat this, the Australian Medical Association has called on all jurisdictions to amend the Health Practitioner Regulation National Law, exempting mandatory reporting of doctors experiencing mental illness, similar to the amended model successfully used in Western Australia [51]. For senior doctors, mental health stigma can have severe implications on their ability to work, further exacerbating financial strain during the pandemic, as well as threatening workforce retention. This is further supported by a qualitative study exploring views of senior doctors on mental illness in healthcare in the UK, which also described concerns regarding privacy and confidentiality, and negative career implications [52]. Thus, solutions to improve mental wellbeing in doctors must include actions to reduce the stigma related to mental illness and improve access to psychological support.

### 4.1. Implications and Future Research

This study documents the significant impacts that the COVID-19 pandemic has had on senior doctors. The risks of moral injury, burnout, and mental health issues in medical leaders within the public and private hospital sectors are important to recognise. The results also highlight important concerns regarding the delivery and quality of patient care, and worsening gender inequality in the workplace during the pandemic. Gender inequality is an important issue in medicine [53] and further research is required to explore its impacts on mental health in senior doctors during the COVID-19 pandemic. Most importantly, our findings emphasise how occupational mental health risk factors can be effectively addressed through appropriate communication, wellbeing support, and the provision of a safe workplace. Senior doctors pointed to the key role they can play as healthcare leaders and clinical experts in future crisis prevention, planning, and management, and the development of policies and guidelines. Strategies that reduce the stigma of mental illness and enhance wellbeing support for all HCWs are essential. The implementation of any new supports and strategies should be supported by research that evaluates the effectiveness of these interventions on mental health. Longitudinal research that documents the continuing psychosocial and workplace impacts of the pandemic on senior doctors is also needed.

### 4.2. Strengths and Limitations

This study highlights the experiences of senior doctors during a pandemic. The large sample size with a breadth of participants from across Australia enabled an in-depth understanding of the challenges, providing insight to inform future intervention strategies. The senior doctors provided a high volume of detailed responses; a testament to the many issues experienced by this group during the pandemic. Most participants resided in Victoria, the Australian state that experienced the greatest impact from the COVID-19 pandemic during the second wave (in 2020) [54]. Data on broader demographic characteristics such as ethnicity or immigrant status were not collected to reduce the length of the survey and preserve anonymity. However, these data are important, as other reports have documented an increased morbidity and mortality in doctors from ethnic minority backgrounds during the COVID-19 pandemic [55]. The voluntary nature of the survey may have resulted in selection bias and response bias. Respondents experiencing mental health issues may have been more likely to participate in the survey, leading to overreporting. However, this study resonates strongly with the findings from other Australian [20,56,57] and international [21,22] HCW studies.

## 5. Conclusions

The health and wellbeing of the healthcare workforce is vital to the sustainability of the future healthcare system. This study is one of the first studies reporting the significant impact of COVID-19 pandemic on senior doctors. Better leadership and planning at organisational and government levels, adequate HCW representation, proactive support for mental wellbeing, and reduction in mental health stigma are crucial to ensure workforce retention and reduce negative effects on patient care during future crisis events.

## Figures and Tables

**Figure 1 ijerph-19-03079-f001:**
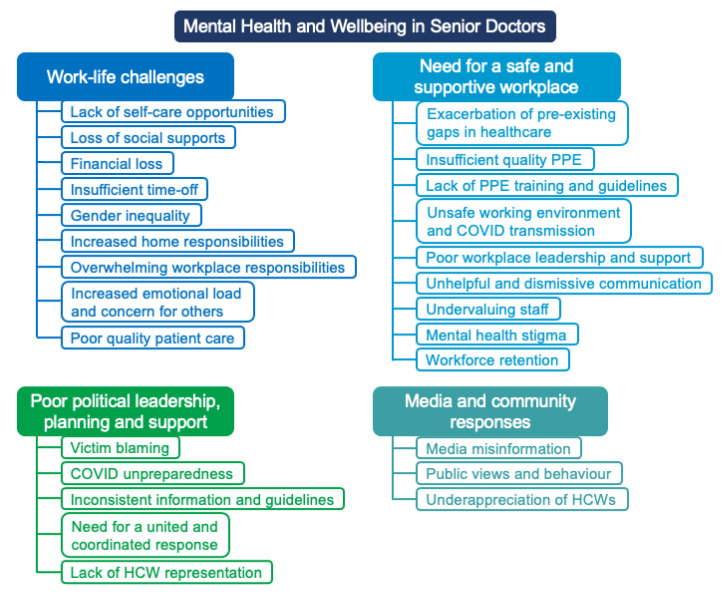
Themes: Mental health and wellbeing of Australian senior doctors during the COVID-19 pandemic.

**Table 1 ijerph-19-03079-t001:** Free-text survey questions.

Free-Text Questions	Number of Senior Doctor Responses
What do you think would help you most in dealing with stress, anxieties, and other mental health issues (including burnout) related to the COVID-19 pandemic?	869
2. What did you find to be the main challenge you faced during the COVID-19 pandemic?	1032
3. What strategies might be helpful to assist frontline healthcare workers during future crisis events like pandemics, disasters, etc?	852
4. Is there anything else that you would like to tell us about the impact of the COVID-19 pandemic or regarding supports that you feel are useful for wellbeing?	373

**Table 2 ijerph-19-03079-t002:** Participants’ characteristics (*n* = 1278).

Characteristic	Frequency	Percent (%)
Age (years)		
20–30	4	0.3
31–40	349	27.3
41–50	497	38.9
50–64	383	30.0
65–70	34	2.7
71+	11	0.9
Gender		
Female	752	58.8
Male	517	40.5
Non-Binary	5	0.4
Prefer not to say	4	0.3
State		
Victoria	973	76.1
New South Wales	126	9.6
Queensland	58	4.5
Western Australia	44	3.4
South Australia	40	3.1
Tasmania	14	1.1
Norther Territory	13	1.0
Australian Capital Territory	10	0.8
Work Location		
Metropolitan Area	1118	87.5
Regional Area	150	11.7
Remote Area	10	0.8
Frontline Area		
Medical Specialty ^1^	301	23.6
Anaesthetics/Perioperative Care	197	15.4
Emergency Department	182	14.2
General Medicine	103	8.1
Respiratory Medicine	100	7.8
Intensive Care Unit	98	7.7
Surgical Specialty	73	5.7
Aged Care ^2^	64	5.0
Infectious Diseases	58	4.5
Palliative Care	49	3.8
Other ^3^	53	4.1

^1^ Medical specialty include all other medical (i.e., non-surgical) sub-specialties apart from general medicine, respiratory medicine, aged care, infectious diseases, and palliative care, which are reported separately in this table due to their potentially higher risk of exposure to COVID-19. ^2^ Aged Care = Hospital Aged Care: 59 and Residential/Non-Hospital Aged Care: 5. ^3^ Other = hospital outreach clinics: 27, Radiology: 8, Pathology: 6, Leadership Role: 3, or other role: 9.

**Table 3 ijerph-19-03079-t003:** Work–life challenges.

Sub-Theme	Quotes
Lack of self-care opportunities	“[For mental health I need] time to exercise, meditate, cook etc. rather than being busier by working full time while homeschooling also”. *(Female, Age 41–50, Medical Specialty, Q1)* “[For mental health I need] better sleep, more exercise, eating better. Developing new interests. Limited by inertia and lack of energy—burnout/anxiety”. (*Female, Age 41–50, Aged Care, Q1*)
Loss of social supports	“Human contact is important and impossible. I haven’t touched a person in over two months without a latex glove, living alone and single, not being able to see friends. That is hard, particularly after a long day in COVID-affected nursing homes with multiple deaths and traumatic scenes”. *(Male, Age 31–40, Aged Care, Q1)*
Financial loss	“Initially the biggest stress was the lack of telehealth item numbers which dropped income to ⅓ of usual values, despite hours tripling”. *(Female, Age 50–64, Other Medical Specialty Area, Q1)* “Financially, this has been a strain with my private practice income dropping by 60–70%. I am unsure when that is going to return to normal which is quite stressful as we have financial commitments and I do not want to lay off any of my staff”. *(Male, Age 41–50, Respiratory Medicine, Q1)*
Insufficient time-off	“The cumulative stress is more than I think people/systems realise, and that is why I am seeing so many staff leaving or having to take extended leave because they have hit the wall at the 6–9 month mark. This is a marathon—not a sprint, and most of us are not elite athletes with the psychology that goes with it…Maybe in times like these, all frontline workers should be given extra leave, so they are not afraid to use their leave when they need it, rather than “saving it up” for afterwards”. *(Female, Age 41–50, Palliative Care, Q3)*
Gender inequality	“There is a pervading attitude that a) women will deal with all the child rearing aspects and b) this means they are now fairly useless from an employment perspective, and this will hurt their careers in ways that they cannot recover from. This will end up true as a self-fulfilling prophecy without better recognition of the impact of COVID-19 on health workers who are mothers and policy and financial steps taken to change this- otherwise we will see worsening of the current economic and senior leadership inequality between women and men”. *(Female, Age 41–50, Medical Specialty, Q4)*
Increased home responsibilities	“Having a caring role for my young (pre-school) children and also my elderly mother is even more challenging during the pandemic, and I feel that my (older, male) colleagues have little understanding of how challenging this is. The pandemic has amplified sexism in the workplace”. *(Female, Age 31–40, Intensive Care Unit, Q4)* “The Victorian government have made it difficult to send children to school and childcare. Schools and childcare have then passed this difficulty onto parents…I have been asked to complete homeschooling exercises despite my child being at school. I have been asked to be keeping track of the online learning timetable in order to remind the school supervisors of my child’s classes. I have also been asked to send in a signed form every week indicating my child is attending in the next week. The requirements of school and childcare have been unnecessarily onerous for a frontline health worker”. *(Female, Age 41–50, Medical Specialty, Q1)*
Overwhelming workplace responsibilities	“[Main challenge was] increased workload both clinical and non-clinical. Overtime +++. High rates of sick leave of medical staff with no availability of extra staff. Working with limited staffing on every shift creating fatigue and burnout”. *(Female, Age 41–50, Emergency Department, Q2)* “[Main challenge was] maintaining momentum and energy to deal with constant changes (e.g., DHHS guidelines, sudden changes in testing requirements overnight) and keeping up with clinical knowledge and data”. *(Female, Age 41–50, Infectious Diseases, Q2)*
Increased emotional load and concern for others	“There’s never been a more challenging time in my experience as a paediatrician. Managing patients, checking in on my staff’s mental health, managing my family with my husband unemployed and one son still in senior school. I feel I’m not looking out for me as there’s nothing left in the tank!” *(Female, Age 50–64, Medical Specialty, Q4)* “[Main challenge was] holding the fear and anger of all my staff. Managing my own fears for their safety. Fears that we would be unable to provide basic care to people. Fear of finishing it all with PTSD”. *(Male, Age 41–50, Emergency Department, Q2)* “I am a co-director of training in Victoria and I really worry about the impact of COVID on trainees/registrars. Our registrars are working in our ‘resp zone’ with support/supervision available as needed—but not with constant support…We keep hearing comments from them [the junior doctors]: ‘I’m the lamb to the slaughter tonight am I’ and ‘if a trainee died from COVID would anyone even care?’—which is extremely worrying…” *(Female, Age 31–40, Emergency Department, Q4)* “I have more concern for junior medical staff. They have borne the brunt of the difficult clinical work, long hours, and roster changes. And they are still expected to study and progress in their career despite enormous uncertainty”. *(Female, Age 41–50,* *Medical Specialty, Q4)*
Quality of patient care	“The main challenge though was the psychological impact of having so many sick patients simultaneously and trying to communicate remotely with families so that they understood what was happening with their relatives. Rationing of treatment because of a lack of ventilators and ECMO meant that clinicians were forced to make very difficult moral decisions. I have never witnessed so many people dying in such a short space of time and they all died alone”. *(Male, Age 31–40, Respiratory Medicine, Q2)* “One of the clinics I work at has decided to be completely hidden from patients and do telehealth 100% of the time. My main challenge is to try to justify to my paediatric patients (and my conscience) why providing inferior medical care to children is meant to be in their best interest (given that these medical conditions will affect them more than the COVID)”. *(Female, Age 41–50, Medical Specialty, Q2)*

Questions are provided in Table 1; DHHS = department of health and human services; PTSD = post-traumatic stress disorder; resp zone = working with COVID patients; ECMO = extracorporeal membrane oxygenation.

**Table 4 ijerph-19-03079-t004:** Poor workplace safety, support, and culture.

Sub-Theme	Quotes
Exacerbation of pre-existing gaps in healthcare	“I think the pandemic has merely exposed the pre-existing issues we were struggling with. Colleagues who are highly supportive were even more so, those who are often draining or difficult in teams deteriorated. Shortages of staff and resources including PPE worsened. Lack of leave worsened. The drain on limited services to provide education to junior staff and students worsened due to increased clinical load. The only resource we had we could increase was personal sacrifice of our own wellbeing and the systems don’t seem to have acknowledged the pre-existing nature of these issues”. *(Female, Age 41–50, Palliative Care, Q4)*
Insufficient quality PPE	“[Main challenge was] lack of appropriately fitted PPE—not enough N95 masks and no fit testing which should be mandated. I have no confidence that the N95 masks fit me properly. I don’t think my organisation was totally honest regarding PPE…We were told no surgical masks if looking after “low risk patients” on coronary care and surgical mask only for looking after COVID patients who were not coughing and not having an aerosolizing procedure. I gather these guidelines have now been changed due to the high numbers of health care worker infections—this was obvious from the beginning and has made it hard to trust hospital administration. There has been no attempt to get us proper fit testing—this means we are working in an unsafe workplace”. *(Female, Age 50–64, Medical Specialty, Q2)*
Lack of PPE training and guidelines	“I also don’t appreciate the lag in recommending N95 for all workers looking after suspected and confirmed COVID cases. I have had some PPE training but no fit testing. And the training I had was for a duck billed N95 at the start of the pandemic. Now I’m being given rigid 3M masks with no training other than being sent a link to an instructional video. All of this needs to be better!” *(Female, Age 41–50, General Medicine, Q3)*
Unsafe working environment and COVID transmission	“The hospital is very poorly set up for cross contamination prevention especially opening of doors, cleaning of common areas, hot desking. Inability to work from home due to abysmal IT and lack of digital pathology. Staff do not practice social distancing during meal and coffee breaks”. *(Male, Age 50–64, Other, Q2)*
Poor workplace leadership and support	“Executive staff are invisible as mostly working from home. They only look at the numbers, not the complexities of patients”. *(Female, Age 50–64, Emergency Department, Q2)* “I work in private practice and a public hospital. The hospital asked where I’d work but never how they’d help if I got sick. Feel unsupported and excluded as a contractor. My organisation does not even pay for meetings between departments”. *(Female, Age 41–50, Medical Specialty, Q2)* “My GP gave me permission to take time off from work after I had recovered from COVID. I did not understand the emotional toll that having COVID would take on me even though I was not unwell. I was very angry and traumatized when I returned to work. A return-to-work plan and emotional support via my public health employer would have been good. Recognition that I did contract the illness at work instead of pretending it didn’t happen. Acknowledgement, support and an apology for not instituting safe COVID practices is what I hoped for and never came”. *(Female, Age 41–50, Surgical Speciality, Q1)*
Unhelpful and dismissive communication	“Managers don’t take doctors’ risks seriously. Too busy managing up to the people above them. They were slow in putting in new policies, ignored doctors’ concerns, minimised the support for those who could work from home, didn’t get fit testing, discouraged the use of PPE in the first wave, stopped people bringing in their own PPE- as it would raise concerns/anxiety in others. So, what would help—serious focus on listening and enacting changes in relation to doctors’ concerns, management accountability in real time, honesty that they don’t have all the answers or protective equipment available if this is the case. Not expecting and demanding business as usual in terms of efficiency and numbers. A caring culture that is inclusive”. *(Female, Age 50–64, Medical Specialty, Q1)*
Undervaluing staff	“[Main challenge was] the lack of care and support the hospital actually showed to the doctors and nurses working on the COVID ward. I was bullied and so were many of the staff on the COVID ward into wearing inappropriate PPE. They only changed practice because the hospital had an outbreak and it made it into the media. They showed little concern when the pandemic started for workers and cared little… Having executives and nurses who had climbed the ladder but are not well educated in getting the latest data was a huge failing and one that impacted on every worker’s mental health in our hospital”. *(Male, Age 41–50, General Medicine, Q2)* “Support and commiseration from colleagues, knowing we are all equally undervalued and disposable in our health system”. *(Female, Age 31–40, Medical Specialty, Q4)*
Mental health stigma	“The stigma and negative consequences of reaching out for help should be abolished so we do not have to suffer alone and suffer without formal help and support”. *(Female, Age 41–50, Community Clinic, Q4)* “[Strategies to help] on an individual level—early access to support. Along with this, a more concerted effort from health services to encourage their use and disarm the notion (whether real or not) that accessing mental health services may lead to consequences for future employment”. *(Male, Age 41–50, Anaesthetics/Perioperative Care, Q3)*
Workforce retention	“I am concerned if we don’t have a lot of ongoing genuine professional psychological support, we are going to see a lot of burnout and HCWs leaving the profession”. *(Female, Age 41–50, Intensive Care Unit, Q4)*

Questions are provided in Table 1.

**Table 5 ijerph-19-03079-t005:** Poor political leadership, planning, and support.

Sub-Theme	Quotes
Victim blaming	“[We need] the government to stop making off the cuff stupid remarks that throw HCWs under the bus for a positive diagnosis...More thoughtful and considered public statements by CHO [chief health officer], Health Ministers, Premiers, PM [prime minister] etc that have some nuance and don’t blame HCWs immediately for everything will go a long way to minimize stigma and collateral damage of COVID panic in this country”. *(Male, Age 41–50, Radiology, Q1)* “They [doctors] should not be blamed if they get infected, media is so nasty in this country. They [media] put their camera in front of the hospital for a few days and every one of us going to the hospital felt like criminals because a poor doctor was found positive”. *(Male, Age 41–50, Respiratory Medicine, Q3)*
COVID unpreparedness	“[Main challenges were] a useless CEO [chief executive officer] of our public hospital, a useless department of health in Victoria which is totally dysfunctional, a useless disorganised supply of inadequate PPE, a very delayed response to increasing numbers of community transmission of COVID. Working in a hospital in which 750 staff are furloughed. Dysfunctional work environment at a global hospital level”. *(Female, Age 41–50, Anaesthetics/Perioperative Care, Q2)* “[We need] a predetermined disaster management plan that is actually communicated and implemented in a timely manner, rather than the reliance on a grass roots movement in local groups to plan and manage both patient and HCW safety. The absolute avoidance of politicisation of a disaster response, and a well communicated public education program to assist in preparation and management”. *(Female, Age 31–40, Anaesthetics/Perioperative Care, Q3)*
Inconsistent information and guidelines	“[Main challenge was] the inconsistent, and at times, lacking communication around healthcare organisation policies; often-lagging DHHS (Vic) guidelines for PPE use (for health care workers) and aged care response to outbreaks; the mixed public health messages; frustration with inconsistent public response to pandemic management strategies; lack of accountability of politicians (in Victoria) for pandemic management failures”. *(Male, Age 41–50, Hospital Aged Care, Q2)*
Need for a united and coordinated response	“[We need] a centralized Department of Health in Victoria—one set of guidelines that everyone has to follow and prioritize staff safety and/or WorkSafe involvement in healthcare. Nimble response and willingness to adapt when healthcare workers are at risk. Priorities that allow training and staff development during usual times not just when in crisis, so we are ready”. *(Female, Age 41–50, Anaesthetics/Perioperative Care, Q3)*
Lack of HCW representation	“[We need] improved lines of communication from the top down. Have frontline HCW appropriately represented in working groups making key decisions that affect us” *(Female, Age 41–50, Emergency Department, Q3)*

Questions are provided in Table 1. DHHS (Vic) = department of health and human services in Victoria; WorkSafe = health and safety regulator and manager of workers compensation scheme in Victoria.

**Table 6 ijerph-19-03079-t006:** Media and community responses.

Sub-Theme	Quotes
Media misinformation	“[We need] sensible government and media reporting and less scaremongering and psychological barrage to get people to do what they want”. *(Male, Age 31–40, Anaesthetics/Perioperative Care, Q1)*
Public views and behaviour	“I wish I could change the behaviour of people who obviously do not care about the effects of their action by not following public health directives, or who are actively protesting against the need to socially distance, wear masks and avoid crowds”. *(Female, Age 41–50, Emergency Department, Q4)*
Underappreciation of HCWs	“I feel that many of my colleagues’ efforts are under appreciated by the general community. They are putting their health at risk while some sectors of the community are debating the need for masks and isolation. It is the health care workers who pay the highest cost when the community fails to control the virus”. *(Male, Age 50–64, Radiology, Q4)*

Questions are provided in Table 1.

## Data Availability

Data available upon reasonable request to the corresponding author.

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
