# Peer review of "The Workplace and Psychosocial Experiences of Australian Senior Doctors during the COVID-19 Pandemic: A Qualitative Study"

_ijerph, 2022, doi:10.3390/ijerph19053079_

Round 1

Reviewer 1 Report

Dear Authors, 

Thank you for this important work on the lived experience of Australian Doctors during the pandemic. You have raised significant concerns and important aspects for change. 

This is a significant and important piece of work. Although it may raise some controversial concerns regarding the experiences of senior Doctors during covid-19 in Australia. The themes raised with evidence provide insight to the many difficulties faced by the medical profession during the second covid outbreak. The review adds depth to previous research by exploring the qualitative responses of a national survey. This paper adds to the limited knowledge regarding frontline workers lived experience during the pandemic and the challenges faced by senior staff across various clinical settings. This makes a significant contribution to the field of health disaster response in Australia, for the current pandemic and for future planning. The article is clear, relevant and well structured. Data is well presented, and themes are easy to follow. Conclusions are consistent with themes and provide insight for future research.

There are only minor changes suggested: 

line 62, Doctor requires an S 

Lines 118-121, seem to be the instructions for the author this requires editing. 

Kind Regards 

Author Response

Dear Authors, 

Thank you for this important work on the lived experience of Australian Doctors during the pandemic. You have raised significant concerns and important aspects for change. 

This is a significant and important piece of work. Although it may raise some controversial concerns regarding the experiences of senior Doctors during covid-19 in Australia. The themes raised with evidence provide insight to the many difficulties faced by the medical profession during the second covid outbreak. The review adds depth to previous research by exploring the qualitative responses of a national survey. This paper adds to the limited knowledge regarding frontline workers lived experience during the pandemic and the challenges faced by senior staff across various clinical settings. This makes a significant contribution to the field of health disaster response in Australia, for the current pandemic and for future planning. The article is clear, relevant and well structured. Data is well presented, and themes are easy to follow. Conclusions are consistent with themes and provide insight for future research.

Response 1: We thank the reviewer for their positive review and helpful feedback regarding our manuscript.

There are only minor changes suggested: 

line 62, Doctor requires an S 

Lines 118-121, seem to be the instructions for the author this requires editing. 

Response 2: We apologise for these two errors, which have been corrected.

Kind Regards 

Reviewer 2 Report

The topic of workplace experiences and psychosocial factors of doctors in Australia is interesting.   Overall, I found the article enjoyable to read, and it covered comprehensive topics such as work-related stress, frustration, uncertainty, issues related to work-life balance, burnout, financial issues, gender inequality, and so forth. I think that the paper can be strengthened further.  Please see my comments below.  

  1. Line 29 discusses negative societal opinions. Negative societal opinions about what? Perhaps restate this?
  2. Box 4 has a spelling error.  It should be Medical misinformation.
  3. Lines 233 to 234 – other research around the world and in commonwealth countries also shows the psychosocial impacts and social determinants of health issues from COVID-19 on frontline healthcare workers, perhaps worth citing, See https://www.researchgate.net/profile/Iffath-Unissa-Syed/publication/344352723_COVID-19_and_Healthcare_Workers'_Struggles_in_Long_Term_Care_Homes/links/5ff8b5a9a6fdccdcb83ed1d4/COVID-19-and-Healthcare-Workers-Struggles-in-Long-Term-Care-Homes.pdf    See also https://bmchealthservres.biomedcentral.com/articles/10.1186/s12913-021-06393-5
  4. Limitations – There are additional limitations that can be discussed. For example, the survey did not collect race/ethnicity or immigrant status data, which is an important omission.  Racialized front line healthcare workers have been experiencing significant morbidity and mortality during the COVID-19 pandemic, a brief discussion in this section could potentially fit. 

If you decide to incorporate these revisions, please upload a manuscript that contains tracked changes or other method to highlight revisions.  Thank you for the opportunity to review this work.

Author Response

The topic of workplace experiences and psychosocial factors of doctors in Australia is interesting.   Overall, I found the article enjoyable to read, and it covered comprehensive topics such as work-related stress, frustration, uncertainty, issues related to work-life balance, burnout, financial issues, gender inequality, and so forth. I think that the paper can be strengthened further. 

We thank the reviewer for their positive review and helpful feedback regarding our manuscript.

Please see my comments below.  

1. Line 29 discusses negative societal opinions. Negative societal opinions about what? Perhaps restate this?

Thank you for this helpful suggestion. We agree and have changed the theme title to be clearer: “media and community responses”. We have made this change accordingly throughout the manuscript.

2. Box 4 has a spelling error.  It should be Medical misinformation.

We apologise for this spelling error. The sub-theme should read “media misinformation” and have made the change in Box 4.

3. Lines 233 to 234 – other research around the world and in commonwealth countries also shows the psychosocial impacts and social determinants of health issues from COVID-19 on frontline healthcare workers, perhaps worth citing, See https://www.researchgate.net/profile/Iffath-Unissa-Syed/publication/344352723_COVID-19_and_Healthcare_Workers'_Struggles_in_Long_Term_Care_Homes/links/5ff8b5a9a6fdccdcb83ed1d4/COVID-19-and-Healthcare-Workers-Struggles-in-Long-Term-Care-Homes.pdf    See also https://bmchealthservres.biomedcentral.com/articles/10.1186/s12913-021-06393-5

Thank you for this interesting comment and the suggestions to incorporate other research from commonwealth countries. We note that the article by Syed and Ahmad (2020) is a review of issues related to healthcare workers in long term care homes. While an interesting paper, this covers a different population of healthcare workers and so is not relevant to our discussion, which focused on the impacts of the COVID-19 pandemic on senior doctors. As such, we politely decline the suggestion to cite the paper. However, we feel that the qualitative study on critical care physicians in Canada by Leigh et al. (2021) fits well in our discussion and have cited this in our manuscript on line 242.

4. Limitations – There are additional limitations that can be discussed. For example, the survey did not collect race/ethnicity or immigrant status data, which is an important omission.  Racialized front line healthcare workers have been experiencing significant morbidity and mortality during the COVID-19 pandemic, a brief discussion in this section could potentially fit. 

Thank you for this feedback. We have added further limitations regarding demographic characteristics including ethnicity and immigrant status on lines 332-336.

If you decide to incorporate these revisions, please upload a manuscript that contains tracked changes or other method to highlight revisions.  Thank you for the opportunity to review this work.

Reviewer 3 Report

This is well written and the points are highly relevant not only in Australia but in many other countries as well

Author Response

This is well written and the points are highly relevant not only in Australia but in many other countries as well

We thank the reviewer for their positive review and thoughtful feedback regarding our manuscript.

Round 2

Reviewer 2 Report

The topic of workplace experiences and psychosocial factors of doctors in Australia is interesting.   A number of suggestions were made in the previous review; however, the author(s) failed to implement some of them; which weaken the manuscript. The suggestions are summarized below, including some new and additional remarks.  

  1. Line 49 – This is vague. What kind of altered work practices?  Perhaps consider a sub-theme which is found in Box 1, 2, 3, or 4? Also, what kind of role changes? You’ve mentioned the roles (clinicians, educators, leaders) in lines 64-65, but these roles nor their changes were recorded e.g. sub-themes in Box 1, 2, 3, 4.
  2. Line 78 – what kind of associations? Professional associations? Professional regulatory bodies?
  3. Section 2.2 - You’ve defined senior doctors with particular post-graduate specialist training, but consultants are an occupational category, they do not necessarily require further educational training.  People with a bachelors degree in business administration can be consultants too; thus, this needs to be separated out or expanded upon.  For staff specialists (and senior doctors in general), was there an average number of years of post-graduate training that the doctors received?
  4. Line 96 - this statement is a bit misleading because you only mention general practitioners, but you’ve stated in Table 1 that other types of doctors were also excluded, such as palliative care physicians, respiratory care, infectious disease, and aged care, but have not stated those here. Perhaps include a qualifier at the end of this sentence e.g. “among others”
  5. Line 145 – You’ve crossed out negative societal opinions and replaced it with media and community responses, but figure 1 still has negative societal opinions and it repeats the exact things. Was this part of the tracked changes and the old figure was kept for comparison/contrast?
  6. Lines 238 – Strangely you’ve mentioned the general adult population as a comparison group regarding the effects on daily life, but not other front-line occupations, nor the social determinants of health (which were previously suggested).  Consider including a brief discussion on social determinants, and those intersecting impacts from COVID-19 on frontline healthcare workers.
  7. The authors have not cited some of the sources suggested from the previous review. In addition to those, See also: Canadian clinicians adopting the virtual health care during COVID-19 pandemic, found at https://concurrentdisorders.ca/download/1803/  and also https://www.liebertpub.com/doi/abs/10.1089/152460900318687
  8. Line 351 – was consent provided in writing? Please state clearly here.

If you decide to incorporate these revisions, please upload a manuscript that contains tracked changes or other method to highlight revisions.  Thank you once again for the opportunity to review this work.

Round 3

Reviewer 2 Report

Now suitable for publication.